# Mapping *Opuntia stricta* in the Arid and Semi-Arid Environment of Kenya Using Sentinel-2 Imagery and Ensemble Machine Learning Classifiers

James M. Muthoka [1,*], Edward E. Salakpi [2], Edward Ouko [3], Zhuang-Fang Yi [4], Alexander S. Antonarakis [1] and Pedram Rowhani [1]

1   Department of Geography, University of Sussex, Brighton BN1 9QJ, UK;
    A.Antonarakis@sussex.ac.uk (A.S.A.); p.rowhani@sussex.ac.uk (P.R.)
2   Department of Physics and Astronomy, University of Sussex, Brighton BN1 9QH, UK; E.Salakpi@sussex.ac.uk
3   Regional Centre for Mapping of Resources for Development Technical Services, Nairobi 00618, Kenya;
    eouko@rcmrd.org
4   Development Seed, Washington, DC 20001, USA; nana@developmentseed.org
*   Correspondence: j.muthoka@sussex.ac.uk

**Abstract:** Globally, grassland biomes form one of the largest terrestrial covers and present critical social–ecological benefits. In Kenya, Arid and Semi-arid Lands (ASAL) occupy 80% of the landscape and are critical for the livelihoods of millions of pastoralists. However, they have been invaded by Invasive Plant Species (IPS) thereby compromising their ecosystem functionality. *Opuntia stricta*, a well-known IPS, has invaded the ASAL in Kenya and poses a threat to pastoralism, leading to livestock mortality and land degradation. Thus, identification and detailed estimation of its cover is essential for drawing an effective management strategy. The study aimed at utilizing the Sentinel-2 multispectral sensor to detect *Opuntia stricta* in a heterogeneous ASAL in Laikipia County, using ensemble machine learning classifiers. To illustrate the potential of Sentinel-2, the detection of *Opuntia stricta* was based on only the spectral bands as well as in combination with vegetation and topographic indices using Extreme Gradient Boost (XGBoost) and Random Forest (RF) classifiers to detect the abundance. Study results showed that the overall accuracies of Sentinel 2 spectral bands were 80% and 84.4%, while that of combined spectral bands, vegetation, and topographic indices was 89.2% and 92.4% for XGBoost and RF classifiers, respectively. The inclusion of topographic indices that enhance characterization of biological processes, and vegetation indices that minimize the influence of soil and the effects of atmosphere, contributed by improving the accuracy of the classification. Qualitatively, *Opuntia stricta* spatially was found along river banks, flood plains, and near settlements but limited in forested areas. Our results demonstrated the potential of Sentinel-2 multispectral sensors to effectively detect and map *Opuntia stricta* in a complex heterogeneous ASAL, which can support conservation and rangeland management policies that aim to map and list threatened areas, and conserve the biodiversity and productivity of rangeland ecosystems.

**Keywords:** invasive plant species; remote sensing; extreme gradient boost; random forest; spectral indices; topographic indices

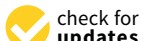



## 1. Introduction

   Globally, grassland biomes form one of the largest covers of terrestrial ecosystems and play a critical role ecologically, socially, and economically. Ecologically, grasslands provide services for carbon sinks [1,2], biodiversity conservation [3], soil conservation [4,5], and forage biomass for herbivores [6,7]. Socially, they provide a livelihood to 800 million people [8,9] and their products contribute significantly to calories and protein consumed globally [10]. Economically, tourism services are found in these biomes and coupled with the livestock market, directly and indirectly, earn governments billions of dollars

in revenue collection and the demand for such services and products is estimated to increase by 2050 [11]. However, while there are many benefits in the conservation of these biomes, they are faced with an increasing number of challenges, which include droughts [12], land use and land cover change [13], land degradation due to overgrazing, deforestation for fuelwood production [14], and the naturalization of Invasive Plant Species (IPS) that have continuously threatened the existence of these habitats and the integrity of these ecosystems [15].

Invasive plant species are a group of plants that naturally thrive in colonized grounds other than their native grounds [16,17]. In the grasslands of Eastern Africa, IPS have found their pathways through deliberate efforts aimed at conservation and the need to create aesthetics in the landscape [18,19]. Unfortunately, the intended benefits of their introduction have been outweighed by their adverse impacts on the ecosystems [20], which threaten the livelihood of the pastoralist communities that rely on them [21,22]. Indeed, IPS tend to reduce land productivity and hinder livestock mobility by occupying areas necessary for palatable pastures [20]. Furthermore, there have been increases in livestock mortality through the ingestion of the IPS fruits, leading to a loss of income [23]. In Kenya, *Prosopis* spp. and *Opuntia stricta* have substantially degraded large areas of pastures and impacted the livelihood of pastoralist communities [24–26]. For example, previous research has identified *Opuntia stricta* as abundant in Laikipia [27,28], negatively affecting rangeland condition and economic losses in excess of US$500 in 48% of household in Laikipia [28]. Due to these challenges, *Opuntia stricta* has grabbed the attention of scientists and decision-makers due to the community outcry of the IPS predicament and need to experiment control measures [29].

*Opuntia stricta* is classified as a Cactus genera and is one of the most invasive plant species causing lots of disturbances in Kenya [20,29,30]. It's a perennial shrub characterized by its red-purple fruits, green cladodes and can attain a height of up to 3.5 metres [23,31]. *Opuntia stricta* is a native plant in America (North, South and Central) [32] but has recently spread in the Kenyan rangelands. It is believed that it entered the country as an ornamental plant with its origin traced to a small town called Dol dol in Laikipia county [33]. It then dispersed to other parts of the country and is now believed to have naturalized in arid and semi-arid lands (ASALs) [19,29]. Its ability to naturalize is attributed to high reproduction through fast growth and the production of seeds that are tolerant of this harsh environment in the ASALs.

*Opuntia stricta* also produce fruits that are palatable to both livestock and wild animals, hence encouraging the dispersion through animal droppings [33,34]. Consequently, the significance of *Opuntia stricta* has been a long-standing rivalry between different schools of thought [20]. While there are some benefits of this IPS, e.g., forages, conservation of soil, water, and some cases known to have medicinal value [35–37], *Opuntia stricta* remains a threat in pastoral communities due to livestock mortality and grassland biome degradation.

Reducing the vulnerability of pastoral livelihood from *Opuntia stricta* requires sound control measures. The management of IPS has employed traditional measures such as manual or mechanical removal of the species and biological controls [38]. For instance, biological control of *Opuntia ficus-indica* in South Africa and Madagascar resulted in the reduction of its densities [39,40]. Additionally, biological control measures have been widely observed to reduce the cost of management as a result of the minimum personnel requirement. In Laikipia, the biocontrol measures have been rolled out on a trial basis with promising results of reduction in physical plant growth and products [41]. However, the long-term management of this IPS requires good knowledge of its spatial extent [29]. Currently, this information is gathered through costly and time-consuming field-based methods [42].

Satellite-based earth observation imagery provides a cost-effective approach to accurately map IPS over large areas to inform management practices. Several studies have identified ways to accurately map IPS from space using multispectral [43] or hyperspectral data [44,45]. Mapping of IPS has often relied on the differentiation of spectral reflectances against the surrounding vegetation [46] and observation of the continuous plants' traits

such as leaf phenology. For instance, Matongera et al. [43] focused on the spectral characteristic of World-View 2 and Landsat imagery to detect bracken fern weeds an IPS in KwaZulu-Natal, South Africa. Earlier efforts to use satellite images have either been limited by poor spatial and spectral resolution of multispectral data or the cost, high dimensionality, availability, and complex processing especially with hyperspectral data [47]. With the spatial and spectral challenges continuously being addressed by the launch of new satellite missions that provide data freely, only computational resources would be needed.

Recent advances in computational power and storage in addition to the use of machine learning algorithms have provided an efficient way to detect and map the IPS. For example, Random Forest and Extreme Gradient Boost algorithms provided high accuracy of (92% and 88%, respectively) in the mapping of the fractional cover of *Prosopis juliflora* an IPS in the Afar Region of Ethiopia using multispectral Landsat 8 data [48]. Similarly, using Sentinel-2 vegetation indices and Support Vector Machines, an accuracy of 80% was achieved while mapping *Rubus cuneifolius* IPS in the KwaZulu-Natal province of South Africa [49]. The technical developments of the new generation satellites such as Sentinel-2 with enhanced spatial, spectral, and temporal resolution provides an opportunity to monitor the spread and undertake detailed landscape mapping of *Opuntia stricta*.

Characterization of these ASAL landscapes is vital for the management of invasive *Opuntia stricta* species. Additionally, knowledge gaps exist especially in overcoming limitations of mapping and invasion science to have methodological research for IPS detection [50]. Therefore, the main aim of this paper was to map *Opuntia stricta* over a heterogeneous ASAL landscape in Laikipia county, Kenya, using ensemble machine learning classifiers (i.e., Extreme Gradient Boost (XGBoost) and Random Forest (RF)) applied to Sentinel-2 data. The classifiers were chosen because of high classification results and the ability to avoid overfitting. Furthermore, the classifiers will test the ability of Sentinel-2 bands, as well as vegetation and topographic indices in accurately classifying *Opuntia stricta*.

## 2. Materials and Methods

The study design is shown in (Figure 1) where we take a four-step process approach. Firstly, we collect all the required data, then we pre-process the data, followed by the training and evaluation of the model. Finally, we create the cover maps from both classifiers and an *Opuntia stricta* mask from the best performing classifier.

### 2.1. Description of the Study Area

In Kenya, ASALs cover 89% of the land mass, and host 36% of the population. Additionally, precipitation per annum ranges between 150 mm–550 mm and 550 mm–850 mm in arid and semi-arid zones, respectively, with temperature and evatranspiration remaining throughout the year [51]. Laikipia is one of the 29 counties in the ASAL region of Kenya and covers an area of about 9500 km$^2$ [52]. The county straddles the equator and is surrounded in the south by Mt Kenya and the Nyandarua ranges while in the north it stretches to more arid plains. Laikipia receives an annual average precipitation of 450 mm over two main rainy seasons, though rainfall can be erratic and varies strongly across the county. This variability of extremely dry or wet events has partly been attributed to the degradation of the environment [53] and is evident across the four Agro-ecological zones (upper highland-sub humid, low highlands, upper midlands, and lower midlands). The study area, which falls among the latter, is notable for ranching, (private and community) beef cattle, and sisal farming due to its non-arable land [26]. It has an overall population of 518,560 [54] who mainly rely on subsistence, rainfed agriculture, and raising livestock in a mixed farming environment. This study will focus on community group ranches in northern Laikipia, namely, Morupusi, Kurikuri, Mukurian, and Ilpolei (Figure 2) which is where the town of Dol Dol is located and is known as the origin of the IPS. Additionally, the location is where we have a nexus of private and communal conservancies which are important to understand for the spatial distribution of *Opuntia stricta* and the pressures it faces.

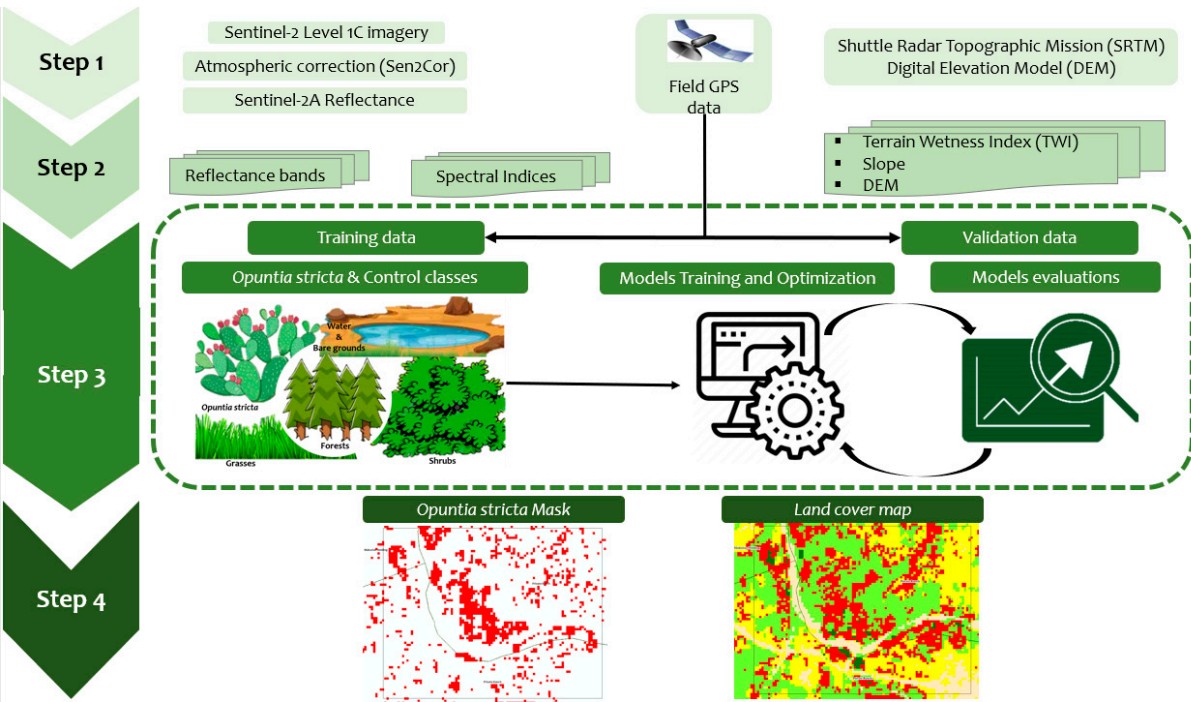

**Figure 1.** Methodology flow chart.

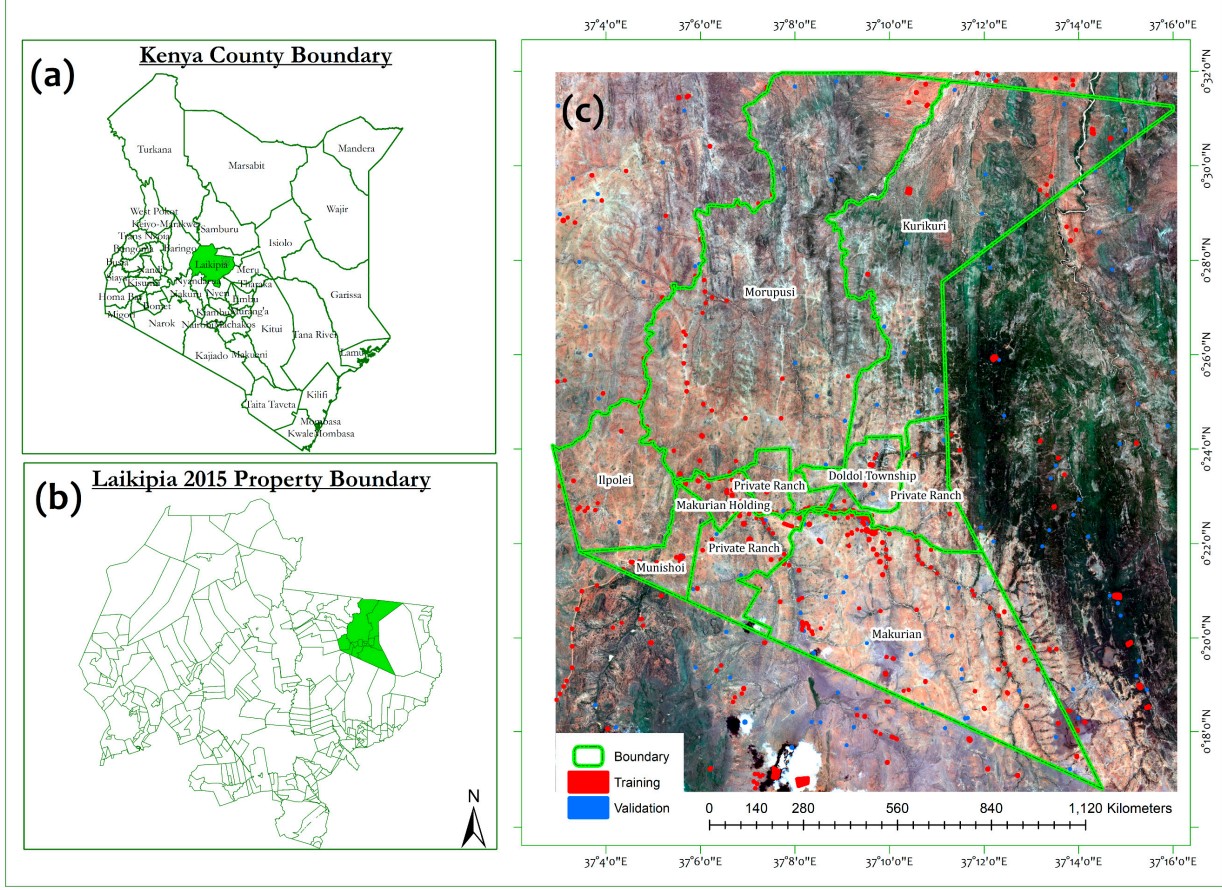

**Figure 2.** A map showing the study area (panel **c**), relative to Laikipia county (panel **b**) as well as the location of Laikipia county in Kenya (panel **a**). Additionally, we show training (red) and validation (blue) data points. The background image is of Sentinel 2a acquired 16 September 2019.

The pressures on the land are driven by land-use and land-tenure issues, resulting from the colonial and post-colonial era [55]. Recently, irregular land allocation [56,57], foreign land acquisition [58], and human–wildlife conflict [55] have led to the privatization of the land which has led to the introduction of hard borders. While the conservancies and private ranches focus on tourism and beef production, the community ranches are used for pastoralism. Pastoralism in the county is practised by the Maasai nomadic community [59] who still claim Laikipia as their ancestral grounds. Additionally, several other pastoralist communities use land that has been abandoned by their owners [60]. The above land adjudication issues have left the pastoralist with limited space and insecurity of tenure, which has often resulted in conflicts and increased land degradation due to overgrazing [58,61]. The poor management of the pastures has also contributed to the spread of the *Opuntia stricta*, which now covers an estimated area of 11,500 ha in the county and is seen as a major environmental challenge impacting the conservation and economic development [62] of the county.

### 2.2. Field Data

Training and testing data to classify *Opuntia stricta* in Laikipia were collected using fieldwork and satellite data. On the ground, two fieldwork campaigns were undertaken to collect training and validation data for several land cover classes. In October 2019, GPS points were collected for *Opuntia stricta*, grasslands, shrublands, and bare ground (Table 1) while during earlier fieldwork, in 2017 and 2018, members from the Regional Center for Mapping of Resources for Development (RCMRD) trained field staff in the region to collect *Opuntia stricta* locations and provide visual estimations of densities by roughly making a count of its bushes and describing the surrounding vegetation type such as grass, shrubs, and trees. For this study, points denoted as moderate to dense with a high percentage canopy cover in the open field were considered. Additional data for the forest, water land cover classes, and clouds and shadows were collected using on-screen digitization of Sentinel-2 data in a false-composite combination as they were visually easy to differentiate as compared to the rest of the classes. These campaigns resulted in 628 points collected for the training and validation of the classification (Figure 2).

**Table 1.** Landcover classes and sample pixel number collected for the classification.

| Classes | Class ID | Description | Samples | |
|---|---|---|---|---|
| | | | Training | Validation |
| Bare ground | 0 | Dry exposed soils | 134 | 18 |
| Clouds | 1 | White material | 2 | 2 |
| Forests | 2 | High-density woody vegetation | 41 | 20 |
| Grasses | 3 | Open and high-density low vegetation | 75 | 38 |
| *Opuntia stricta* | 4 | Target cactus vegetation | 144 | 63 |
| Shadows | 5 | Black material | 1 | 1 |
| Shrubs | 6 | Low lying vegetation | 77 | 20 |
| Water | 7 | Open water | 6 | 3 |

### 2.3. Satellite Image Analysis and Classification

In this section, we discuss the various steps we undertook in image acquisitions, spectral characterization, and analysis, deriving the vegetation and topographic indices, and finally the image classification and accuracy assessment.

2.3.1. Satellite Image Acquisition and Pre-Processing

We used satellite imagery from the Copernicus Sentinel-2 mission, which observes the Earth at 10 m, 20 m, and 60 m spatial resolution and are available every 5–12 days free of cost [63]. Since cloud cover is almost persistent in this region, we selected an image from 16th September 2019 with a cloud cover of less than 10%. The image was available at level 1C hence required additional processing to convert and retrieve the

surface reflectance. Atmospheric and geometric corrections for Sentinel-2 images were performed with Sen2Corv2.8 [64]. Next, we resampled the 20 m resolution bands to get uniform 10 m resolution pixels using the nearest neighbour approach. Here, we used a total of ten spectral bands ranging from the visible to the shortwave infrared wavelengths (Table 2). Furthermore, the extraction of the study area mask was performed for all spectral reflectance bands and data stored for further processing and analysis. All these pre-processing operations were performed within the SNAPv7.0 software [65]. Finally, we included a 30 m resolution Digital Elevation Model (DEM) from the Shuttle Radar Topographic Mission (SRTM) available from the United States Geological Survey (USGS, 2020).

**Table 2.** Sentinel-2 spectral bands.

| Band | Spectral Region | Spatial Resolution (m) | Central Wavelength (nm) |
|---|---|---|---|
| 2 | Blue | 10 | 492.4 |
| 3 | Green | 10 | 559.8 |
| 4 | Red | 10 | 664.6 |
| 5 | Red edge | 20 | 704.1 |
| 6 | Red edge | 20 | 740.5 |
| 7 | Red edge | 20 | 782.8 |
| 8 | Near- infrared | 10 | 832.8 |
| 8A | Near -infrared | 20 | 864.7 |
| 11 | Shortwave Infrared | 20 | 1613.7 |
| 12 | Shortwave Infrared | 20 | 2202.4 |

2.3.2. Spectral Separability

Spectral separability between *Opuntia stricta* and the control classes was assessed using the Jeffries–Matusita (JM) and the Transformed Divergence (TD) methods [66–68]. This was a crucial step in testing how valuable the bands and training pixels are before undertaking further classification analysis. The rationale of the method was due to its ability to evaluate the probability of band pairs to separate between two classes [69]. Shapiro–Wilk normality test was carried out at 95% confidence level to determine if the classes were normally distributed across the spectral bands. Additionally, JM and TD calculate separability by taking an evaluation of the class spectral distances between the mean vectors of the available pairs computed. Here (in)significant distances mean the spectral classes are (less) more separable. These approaches have been widely used for the evaluation of training data sets in land cover classifications [43,70,71]. The JM distance (Equation (1)), and the TD distance (Equation (2)) apply two features, respectively, with an output range of separability values between 0 and 2:

$$J_{xy} = 2\left(1 - e^{-B}\right)\frac{1}{8}(x-y)^t\left(\frac{\Sigma_x + \Sigma_y}{2}\right)^{-1}(x-y) + \frac{1}{2}ln\left(\frac{\left|\frac{\Sigma_x+\Sigma_y}{2}\right|}{|\Sigma_x|^{\frac{1}{2}}|\Sigma_y|^{\frac{1}{2}}}\right) \quad (1)$$

where $x$ and $y$ correspond to first and second spectral signature and $\Sigma_x$ and $\Sigma_y$ are the covariance matrix of sample $x \wedge y$, respectively;

$$TD = 2\left[1 - exp\left(\frac{-D}{8}\right)\right]\frac{1}{2}tr\left[(C_1 - C_2)\left(C_1^{-1} - C_2^{-1}\right)\right] + \frac{1}{2}tr\left[\left(C_1^{-1} - C_2^{-1}\right)(\mu_1 - \mu_2)(\mu_1 - \mu_2)^T\right] \quad (2)$$

where $TD$ is the Transformed Divergence between two classes, $C_1$ is the covariate matrix of class 1, $\mu_1$ is the mean vector of class 1, $tr$ is the matrix trace function, and $T$ is the matrix transposition function. These spectral characterization analyses were performed using ENVI image analysis software v5.4 [72].



### 2.3.3. Extracting Predictor Variables for Classification

Supervised classification requires predictor variables as input data. We used the Sentinel-2 spectral bands (Table 2), vegetation, and topographic indices from the satellite images to improve the model's predictive ability to detect *Opuntia stricta*. Vegetation indices that are suitable to dryland conditions [73,74], reliable for mapping vegetation [75,76], and those that eliminate atmospheric and soil effects [77,78] were included in our analysis (see Supplementary Table S1). To this end, we computed the Ratio Vegetation Index (RVI) [79], Perpendicular Vegetation Index (PVI) [80], Normalized Difference Vegetation Index (NDVI) [81], Infrared Percentage Vegetation Index (IPVI) [82], Atmospherically Resistant Vegetation Index (ARVI) [83], and Modified Secondary Soil adjusted Vegetation Index (MSAVI) [84]. All the computations for the vegetation indices were performed within SNAP.

Additionally, DEM and topographic variables such as slope, aspect [85], and Terrain Wetness Index (TWI) [86], which are known to aid improved decision making for classification tasks, were also included [87–89]. Furthermore, these topographic variables do affect temperature, precipitation, radiation regimes, and moisture demands attributes that indirectly affect vegetation dynamics and microclimates [90]. QGISv3.10 was used to derive the three topographic variables [91]. The formulas used to derive the vegetation and topographic indices are presented in the supplementary materials (Tables S1 and S2, respectively).

### 2.3.4. Classification Algorithms and Accuracy Assessment

Our study employs two ensemble machine learning algorithms, XGBoost and RF, to classify *Opuntia stricta* from satellite imagery. XGBoost has been used extensively for classification tasks due to its ability to develop and weigh multiple decision trees to enhance the overall classification performance [92]. The high prediction skill and accuracies associated with XGBoost can be attributed to its loss function algorithm and how it optimizes weak learners to improve model performance [48,92,93]. The iterative and additive nature of the learning process combined with the use of a strong regularization framework makes the models robust against overfitting [94,95]. The mathematical details of this algorithm are found in Chens' works [92].

The RF algorithm by Breiman [96] classifies by bootstrapping data and creating large numbers of decision trees per bootstrap sample. Each decision tree facilitates decision making by making use of the Classification and Regression Trees (CART) algorithm to split nodes through the reduction in Gini Impurity. Gini impurity measures the probability of a new random variable to be incorrectly classified if it was randomly labelled by the distribution of the training sample. Classification of the data is achieved via bootstrap aggregation. Additionally, we chose this approach based on the results of several recent studies that have demonstrated high accuracy achievements especially for land cover classification assignments [76,93,97,98]. In-depth details of RF are provided by [96] and Strobl works [99].

To build our understanding of the value of adding the additional variables, we first created models with just the spectral bands ((model 1.a for XGBoost and model 1.b for RF) as well as models where we added vegetation and topographic indices to the Sentinel spectral bands (model 2.a for and model 2.b for XGBoost and RF, respectively). Additionally, we analyzed the performance of the models to evaluate the value of adding in these layers.

To ensure we achieved high classification accuracies and avoid model overfitting, we optimized the models by undertaking hyperparameter tuning. The review of literature has shown that numerous approaches exist for hyperparameter tuning [100,101] and do contribute to overall accuracy. For example, Abdi [93] observed an increase in an overall accuracy of 15% in land cover classification while using optimized machine learning. In this study, we used the random search method owing to its ability to perform a comprehensive search and also to ensure that the parameter value search is comparable to other studies [101]. We chose five parameters (Table 3), three for XGBoost and two for RF. The search for the optimum parameters was computed using *k*-fold cross-validation. In

this exercise, hyperparameter tuning was done by using sci-kit-learn's [102] grid search function implemented in the Python programming language.

**Table 3.** Classification hyper-parameters used for XGB and RF algorithm.

| Model | Hyper-Parameter Value | Definition |
|---|---|---|
| XGB | max_depth = 3, learning_rate = 0.1, n_estimators = 100 | maximum depth of a tree to which changes makes the model complex learning rate step size shrinkage used in the updates hence preventing the overfitting maximum number of iterations to the training |
| RF | max_depth = 5, n_estimators = 500 | maximum number of levels for each decision tree tree numbers in the forest |

Finally, we evaluated the ability of the two ensemble machine learning algorithms to discriminate *Opuntia stricta* from other control classes. The accuracy of our classification models was assessed by utilizing the absence and presence model which looks at an error matrix that cross tabulates the absence and predicted patterns versus the observed [103,104]. We generated confusion matrices to compare the true and assigned classes by obtaining the Overall Accuracy (OA), as well as the User Accuracy (UA), Producer Accuracy(PA) [105] and the Kappa coefficient. UA essentially inform us how often a classes on the map will actually be present on the ground, whereas PA gives the probability of a classified class in map being classified as such on the ground. Additionally, due to conceptual issues regarding the Kappa coefficient [106], we computed two additional measures of disagreement between maps, Quantity Disagreement (QD) and Allocation Disagreement (AD) [107]. QD is a measure of the difference in the proportions between the true and predicted classes, while AD measures the difference in the spatial allocations of the actual and predicted land cover classes.

## 3. Results

### 3.1. Analysis of Spectral Separability of Opuntia Stricta and Other Control Classes

The spectral profiles for the various classes are presented in Figure 3. Water, bare ground, and forests show potentially distinct spectral characteristics across the spectral wavelength. However, *Opuntia stricta* and shrubs classes are visually close and they give a broadly similar shape across the spectral wavelength and a new observation is made between grasses and bare grounds.

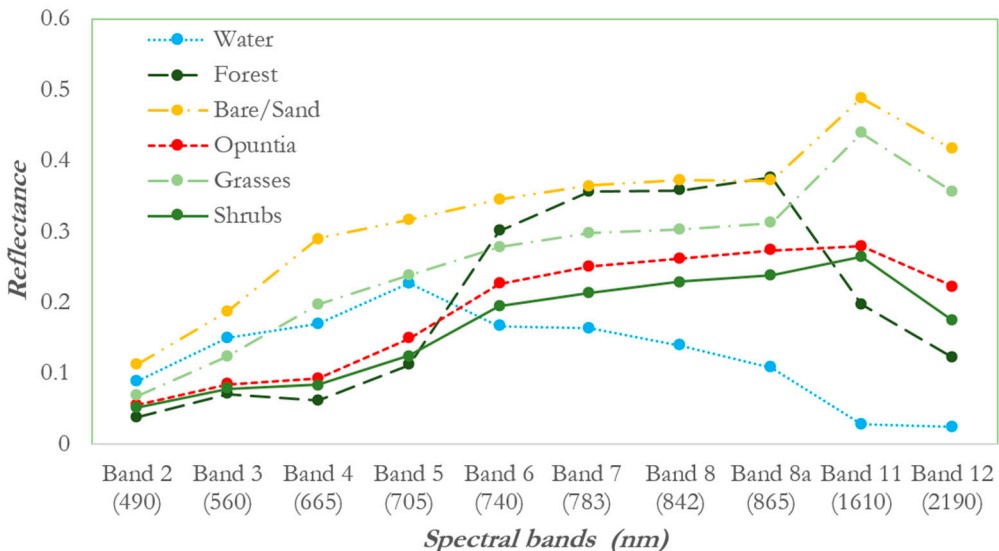

**Figure 3.** Spectral profile of *Opuntia stricta* and control classes.

Results of the Shapiro-Wilk normality test are reported in the supplementary material (Table S3) and the values for JM and TD are found below and above the diagonal, respectively (Table 4). It can be noted that all classes have values greater than 1.27 and 1.59 for JM and TD, respectively, which indicates a good potential for accurate classification. The highest separation was achieved between bare grounds and forests as well as between grasses and forests (1.99 and 2.00 for JM and TD, respectively). Similarly, we also found the weakest separation to be between *Opuntia stricta* and shrubs, as well as between bare grounds and grasses, with values of JM equal to 1.27 and 1.49 while TD values were 1.59 and 1.68, respectively. Finally, our normality test indicates that many of the time, the $p$-value was not less than the significant level of 0.05 which indicate that the land cover classes confirm and follow a normal distribution across the Sentinel-2 spectral bands. The exceptions are bare ground, *Opuntia stricta* and shrubs across NIR, RedEdge1 and Red with $p < 0.05$.

**Table 4.** Separable degree of Jeffries–Matusita (JM: bottom left diagonal) and transformed divergence (TD: upper right diagonal) between Opuntia stricta and other control classes based on Sentinel-2 imagery band.

|  | *Opuntia stricta* | **Shrubs** | **Grasses** | **Bare Grounds** | **Forests** | **Water** |
|---|---|---|---|---|---|---|
| *Opuntia stricta* | - | 1.59 | 1.83 | 1.99 | 1.99 | 2.00 |
| Shrubs | 1.27 | - | 1.89 | 1.99 | 1.99 | 2.00 |
| Grasses | 1.66 | 1.83 | - | 1.68 | 2.00 | 2.00 |
| Bare grounds | 1.92 | 1.95 | 1.49 | - | 2.00 | 2.00 |
| Forests | 1.92 | 1.94 | 1.99 | 1.99 | - | 2.00 |
| Water | 2.00 | 2.00 | 2.00 | 2.00 | 2.00 | - |

### 3.2. Image Classification

The outputs of both classification algorithms (i.e., model 1.a and model 1.b are XGBoost and RF, respectively) with only the Sentinel reflectance bands as input whereas model 2.a and model 2.b are XGBoost and RF, respectively, with vegetation and topographic indices added to the Sentinel-2 reflectance bands and are presented in (Figures 4 and 5). Here, we show that grass and shrubs are the dominant covers, while *Opuntia stricta*, forest, bare ground, and water are less abundant. Grass cover in Figure 4 panels A and B is found mainly to the south, central, and minimal towards the north. Shrubs are found towards the north, central with small patches towards the south of the study area. Additionally, we see a lot of variability in *Opuntia stricta* presence, an indication of co-occurrence of classes. Figure 4 panel A shows the presence of Opuntia stricta towards the north with the south having remnants in patches while Figure 4 panel B shows a co-occurrence of Opuntia stricta with other classes such as grasses and shrubs. Similarly, model 2 results presented in Figure 5 panels A and B portray the spatial spread of Opuntia stricta towards the central area of the map and along the river channels. Shrubs are found more towards the central and north, while the grass is found towards the central and south of the map. The forest cover looks spatially unchanged which may be a result of the good separation between itself and other land cover classes, shown in the JM and TD analysis. An overlay of the land tenure boundaries qualitatively shows that *Opuntia stricta* is present in all the localities with community ranches widely affected. An assessment of the accuracy presents a measure of the models and the variables A and B performance.

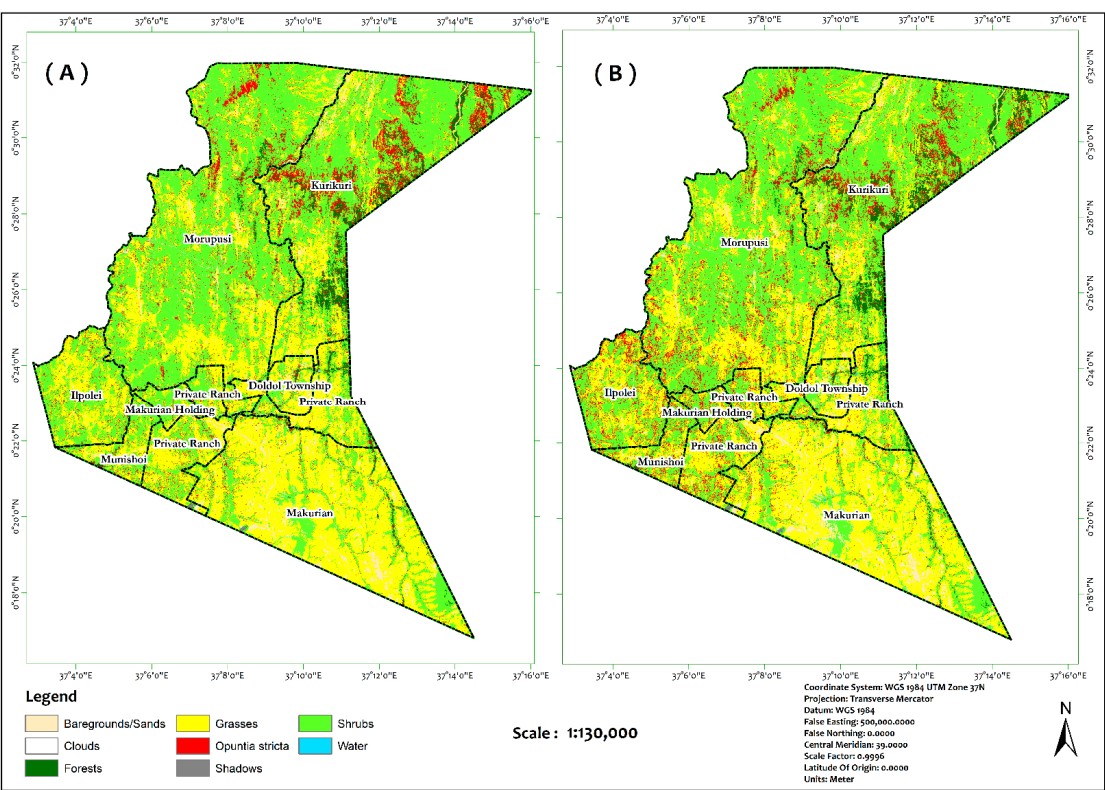

**Figure 4.** Maps showing the classification results of model 1.a and model 1.b (XGBoost and RF, respectively, with only the Sentinel reflectance bands as input), panel (**A**,**B**), respectively. The boundaries present the different land tenures present within the study area.

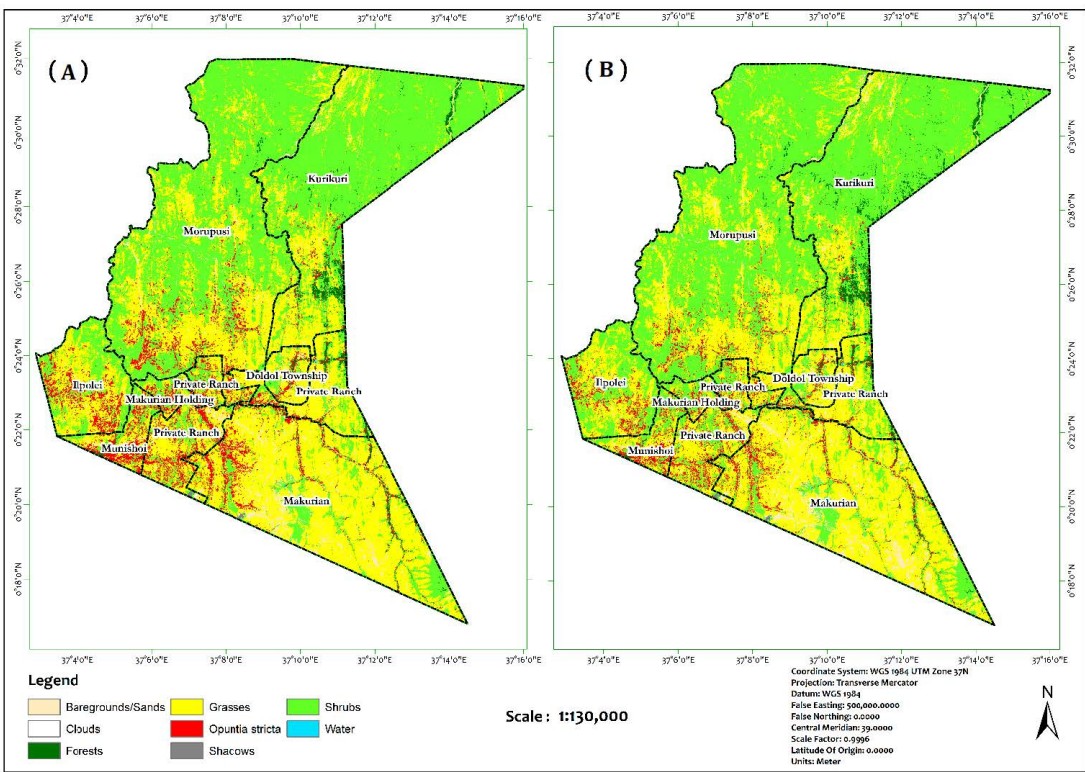

**Figure 5.** Maps showing the classification results of model 2.a and model 2.b are XGBoost and RF, respectively, with vegetation and topographic indices added to the Sentinel-2 reflectance bands, panel (**A**,**B**), respectively. The boundaries present the different land tenures present within the study area.

The overall classification performance of the algorithms based on the predictor variables is given in Tables 5–8. Our results show varied OA levels for both classification algorithms with RF outcompeting XGBoost. Interestingly, model 2 did improve the OA, kappa and reduced the QD and AD for both models. RF achieved an OA of 84.4% and 92.4% for models 1.b and 2.b, respectively, compared to XGBoost 80% and 89.2% for models 1.a and 2.a, indicating an OA increase of 8% and 9% for RF and XGBoost, respectively. These results are reflected as well by the kappa with an improvement of 0.1 and 0.09 for XGBoost and RF, respectively.

**Table 5.** Confusion matrix for model 1.a user's and producer accuracy. The overall weighted accuracy is $0.80 \pm 0.038$.

| Land Cover | | Bare Ground | Cloud | Forest | Grass | Opuntia | Shadows | Shrubs | Water | User Accuracy |
|---|---|---|---|---|---|---|---|---|---|---|
| **Classification Data** | Bare | 24 | 0 | 0 | 3 | 0 | 0 | 0 | 0 | 88.89% |
| | Cloud | 0 | 45 | 0 | 0 | 0 | 0 | 0 | 0 | 100.00% |
| | Forest | 0 | 0 | 23 | 0 | 1 | 0 | 2 | 0 | 88.46% |
| | Grass | 1 | 0 | 0 | 33 | 9 | 0 | 6 | 0 | 67.35% |
| | Opuntia | 0 | 0 | 0 | 2 | 35 | 0 | 3 | 0 | 87.50% |
| | Shadows | 0 | 0 | 0 | 0 | 0 | 29 | 0 | 0 | 100.00% |
| | Shrubs | 1 | 0 | 2 | 0 | 28 | 0 | 31 | 4 | 46.97% |
| | Water | 0 | 0 | 0 | 0 | 0 | 0 | 0 | 32 | 100.00% |
| | Weights | 27 | 45 | 26 | 49 | 40 | 29 | 66 | 32 | |
| | Producer Accuracy | 92.31% | 100.00% | 92.00% | 86.84% | 47.95% | 100.00% | 73.81% | 88.89% | |
| | Overall Accuracy | | | | | | | | | 0.802 |
| | Allocation Disaggrement | | | | | | | | | 0.079 |
| | Quantity Disaggrement | | | | | | | | | 0.117 |
| | Kappa | | | | | | | | | 0.77 |

**Table 6.** Confusion matrix for model 2.a user's and producer accuracy. The overall weighted accuracy is $0.89 \pm 0.031$.

| | | Reference Data | | | | | | | | |
|---|---|---|---|---|---|---|---|---|---|---|
| Land Cover | | Bare Ground | Cloud | Forest | Grass | Opuntia | Shadows | Shrubs | Water | User Accuracy |
| **Classification Data** | Bare | 23 | 0 | 0 | 2 | 0 | 0 | 0 | 0 | 92.00% |
| | Cloud | 0 | 45 | 0 | 0 | 0 | 0 | 0 | 0 | 100.00% |
| | Forest | 0 | 0 | 20 | 0 | 1 | 0 | 0 | 0 | 95.24% |
| | Grass | 3 | 0 | 0 | 34 | 11 | 0 | 6 | 0 | 62.96% |
| | Opuntia | 0 | 0 | 1 | 2 | 57 | 0 | 0 | 0 | 95.00% |
| | Shadows | 0 | 0 | 0 | 0 | 0 | 29 | 0 | 0 | 100.00% |
| | Shrubs | 0 | 0 | 4 | 0 | 4 | 0 | 36 | 0 | 81.82% |
| | Water | 0 | 0 | 0 | 0 | 0 | 0 | 0 | 36 | 100.00% |
| | Weights | 25 | 45 | 21 | 54 | 60 | 29 | 44 | 36 | |
| | Producer Accuracy | 88.46% | 100.00% | 80.00% | 89.47% | 78.08% | 100.00% | 85.71% | 100.00% | |
| | Overall Accuracy | | | | | | | | | 0.891 |
| | Allocation Disaggrement | | | | | | | | | 0.050 |
| | Quantity Disaggrement | | | | | | | | | 0.057 |
| | +Kappa | | | | | | | | | 0.87 |

**Table 7.** Confusion matrix for model 1.b user's and producer accuracy. The overall weighted accuracy is 0.84 ± 0.036.

| | | Reference Data | | | | | | | | User Accuracy |
|---|---|---|---|---|---|---|---|---|---|---|
| **Land Covers** | | **Bare ground** | **Cloud** | **Forest** | **Grass** | **Opuntia** | **Shadows** | **Shrubs** | **Water** | |
| **Classification Data** | Bareground | 26 | 0 | 0 | 5 | 0 | 0 | 0 | 0 | 83.87% |
| | Cloud | 0 | 45 | 0 | 0 | 0 | 0 | 0 | 0 | 100.00% |
| | Forest | 0 | 0 | 21 | 0 | 0 | 0 | 3 | 0 | 87.50% |
| | Grass | 0 | 0 | 0 | 30 | 6 | 0 | 6 | 0 | 71.43% |
| | Opuntia | 0 | 0 | 0 | 1 | 48 | 0 | 3 | 0 | 92.31% |
| | Shadows | 0 | 0 | 0 | 0 | 0 | 29 | 0 | 0 | 100.00% |
| | Shrubs | 0 | 0 | 4 | 2 | 19 | 0 | 30 | 0 | 54.55% |
| | Water | 0 | 0 | 0 | 0 | 0 | 0 | 0 | 36 | 100.00% |
| | Weights | 31 | 45 | 24 | 42 | 52 | 29 | 55 | 36 | |
| | Producer Accuracy | 100.00% | 100.00% | 84.00% | 78.95% | 65.75% | 100.00% | 71.43% | 100.00% | |
| | Overall Accuracy | | | | | | | | | 0.843 |
| | Allocation Disaggrement | | | | | | | | | 0.085 |
| | Quantity Disaggrement | | | | | | | | | 0.070 |
| | Kappa | | | | | | | | | 0.82 |

**Table 8.** Confusion matrix for model 2.b user's and producer accuracy. The overall weighted accuracy is 0.92 ± 0.027.

| | | Reference Data | | | | | | | | User Accuracy |
|---|---|---|---|---|---|---|---|---|---|---|
| **Land Cover** | | **Bare ground** | **Cloud** | **Forest** | **Grass** | **Opuntia** | **Shadows** | **Shrubs** | **Water** | |
| **Classification Data** | Bare | 25 | 0 | 0 | 4 | 0 | 0 | 0 | 0 | 86.21% |
| | Cloud | 0 | 45 | 0 | 0 | 0 | 0 | 0 | 0 | 100.00% |
| | Forest | 0 | 0 | 23 | 0 | 0 | 0 | 1 | 0 | 95.83% |
| | Grass | 1 | 0 | 0 | 34 | 9 | 0 | 3 | 0 | 72.34% |
| | Opuntia | 0 | 0 | 1 | 0 | 60 | 0 | 0 | 0 | 98.36% |
| | Shadows | 0 | 0 | 0 | 0 | 0 | 29 | 0 | 0 | 100.00% |
| | Shrubs | 0 | 0 | 1 | 0 | 4 | 0 | 38 | 0 | 88.37% |
| | Water | 0 | 0 | 0 | 0 | 0 | 0 | 0 | 36 | 100.00% |
| | Weights | 29 | 45 | 24 | 47 | 61 | 29 | 43 | 36 | |
| | Producer Accuracy | 96.15% | 100.00% | 92.00% | 89.47% | 82.19% | 100.00% | 90.48% | 100.00% | |
| | Overall Accuracy | | | | | | | | | 0.923 |
| | Allocation Disaggrement | | | | | | | | | 0.035 |
| | Quantity Disaggrement | | | | | | | | | 0.041 |
| | Kappa | | | | | | | | | 0.91 |

Tables 6 and 8 also gives insight into the contribution of the predictor variables to the classification accuracies. Model 2.b showed improvement in PA and UA scores for the target *Opuntia stricta* of 34.24% and 6.05%, respectively, compared to model 1.b. For the other classes, such as forest, grass, shrubs, and bare ground, the PA increased by approximately, 8%, 11%, 17%, and 2.3% while the UA increased by 8%, 1%, and 41%, respectively. Model 2.a presented mixed results with gains and losses in both PA and UA. Despite the mixed results in PA and UA for model 1, model 2 improved the classification results with both models.

The classification accuracy results of *Opuntia stricta* improved when using model 2 for both classifiers. Our results (Tables 6 and 8) show that *Opuntia stricta* was better classified from the users' perspective as compared to the producers' viewpoint by reaching 98% and 95% for model 2.b and model 2.a, respectively. Additionally, Figure 6 presents the disagreement levels between the two classifiers for models 1 and 2. Firstly, models 1.a and 1.b had the largest overestimation on average by 11.7% and 7% for model 1 and 2,

respectively, which is consistent with our previous result on OA though in reverse since these metrics estimates disagreements between the reference and classified values. Our results show that the use of model 2 substantially reduces the overestimation by 6% and 2.9%, respectively. Model 1.a had the largest QD of all the models. Secondly, our results show that the use of model 2 had the lowest AD with the total disagreement of 7.6% and 10.7% for XGBoost and RF, respectively. Overall, using model 2.b yields the best performance as compared to the rest of the models based on total disagreement though significantly low changes in errors for the same.

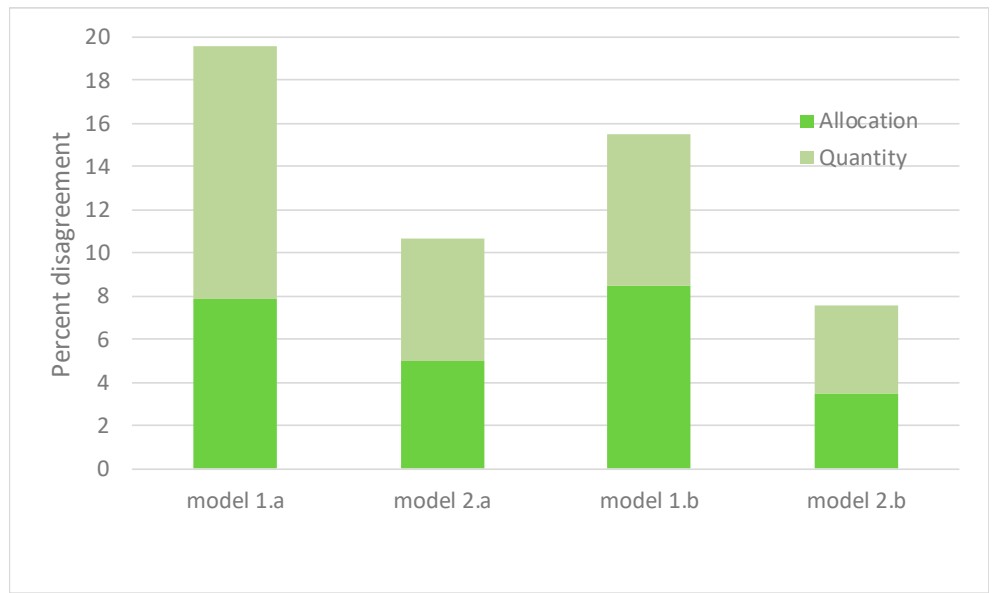

**Figure 6.** Shows the average allocation and quantity disagreements for each model based on the two algorithms. The summation of allocation and quantity disagreements is equal to total disagreement of the models.

Finally, feature importance results are presented in the Supplementary Materials (Figure S1) and varied depending on the model. On one hand, model 2.a feature importance ranked the raw DEM, SWIR1, and slope as the top most important predictor variables with PVI, NDVI, and RVI as the least important for the model. On the other hand, the topographic indices (TWI, Aspect and slope) were the most important predictor variables with red and green bands being the least important for model 2.b. Overall, our results show that topographic indices as compared to other indices largely contributed to the performance of the classifiers for the two models.

## 4. Discussion

Dynamic changes in land cover and poor rangeland management present an opportunity for IPS to spread and have a negative impact on range ecology [108], especially on pastoral lands in the Kenya ASALs. In this study, we have applied two ensemble classifiers to detect and map *Opuntia stricta,* an IPS present in a heterogeneous arid and semi-arid county in Kenya. Our results show that the use of Sentinel 2 spectral bands, vegetation, and topographic indices can characterize heterogeneous ASALs. Here, we first discuss the suitability of the Sentinel-2 spectral bands to map *Opuntia stricta*. Then, we discuss the spatial occurrence of the target class and the control classes while identifying some conceivable ecological implications. Finally, we discuss the accuracy and suitability of the Sentinel-2 data in an upscaled context.

### 4.1. Model Evaluation and Spatial Coverage

In the spectral separability (Figure 3) analysis we provided a qualitative assessment of

the suitability of Sentinel-2 spectral bands to perform classification in these arid environments with abundant *Opuntia stricta* and Figure 6 indicates the spatial occurrence of *Opuntia stricta*. The separability between classes (Table 4) ranged from 1.29–2.0, with the lowest separability between *Opuntia stricta* and shrubs and grasses vs bare ground. Separating species and plant functional types indicates a much finer margin of spectral separation than for broader land cover types such as vegetation vs ground. This indicates that providing separability of similar plant functional types calls for the acceptance of a lower threshold as with the case of Bogan et al. [109]. Chemura and Mutanga [70], and Matongera et al. [43] have also shown that a 1.0 threshold is feasible for the separability analysis.

Spatial coverage of *Opuntia stricta* (Figures 5–7) underscores the ability of Sentinel-2 spectral bands to capture known spatial locations of the IPS, namely in small patches along rivers and settlements as stated in [23,33]. In both these studies, *Opuntia stricta* is reported to have invaded areas near and along rivers, dwellings, and hills which is attributed to the species' seed dispersal mechanism by water, livestock, and wildlife animals [24,30]. Additionally, high land degradation within the community land has contributed to the high invasiveness of *Opuntia stricta* within community conservancies as compared to the private ones which are consistent with previous findings in Laikipia and Samburu [27].

The evaluations of the two classification algorithms in mapping *Opuntia stricta* and the control classes were assessed through absence and presence models and disagreement measures. Our results show that the OA ranged between 80–92% (Tables 5–8) for both classifiers and considering model 1 and 2. Our ability to successfully map *Opuntia stricta* had high overall accuracy for both classifiers which mirrors similar results recorded in previous studies where Mudereri et al. [78] achieved an OA of 87% while mapping an invasive *Striga* weed and those of, Matongera et al. [43] who achieved OA of 80% for detecting invasive *Bracken fern* weed. Both studies are based on comparable sensors (i.e., Sentinel-2 and Landsat-8 images, respectively) though in different ecosystems. Similarly, a recent study has shown that our results are comparable to results obtained through the species distribution model in which Ouko et al. [27] achieved an equally high OA of 97% while modelling invasive *Acacia reficiens* and *Opuntia stricta* in comparable ecosystems in Laikipia and Samburu counties.

Our analysis of disagreement measures revealed that total disagreement was high (Figure 6) when looking at model 1 as compared to model 2. This result is not surprising as both the QD and AD denoted substantial differences in category totals which we attribute to low weights (Tables 5 and 6) and the spatial allocation. However, model 2 lessens these levels of disagreements (Figure 6, Tables 7 and 8), which matches the improvements seen in overall accuracy.

Incorporating vegetation and topographic indices improved the classification accuracies, signaling better separation of the forest, grasses, *Opuntia stricta*, and shrubs and minimizing disagreements. Since the increase in OA happened in both models (XGBoost and RF) signify the general importance of the indices in the classification. This result confirms previous studies that have shown that the addition of indices to the spectral bands results in classification improvements [89]. For example, Matongera et al. [43] achieved a 19.94% improvement in classification accuracy after the addition of vegetation indices; Mudereri et al. [78] had a 1% improvement in detection of *Striga* weed; while Hurskainen et al. [88] achieved an increase of 16.5 percentage points after the addition of auxiliary features which included vegetation and topographic indices in a heterogeneous savanna landscape landcover classification. Reasons for this improvement may be attributed to indices' ability to reduce the influence of soil and atmospheric reflectance in arid and semi-vegetated environments. Similarly, ensemble machine learning algorithms have the ability to inherently combine and learn from multiple variables (i.e., vegetation and topographic indices) as compared to classical classifiers. However, our results contradict the findings of Shoko and Mutanga [110] who achieved the highest OA by using Sentinel-2 spectral bands which they attributed to the optimum positioning of the spectral bands.

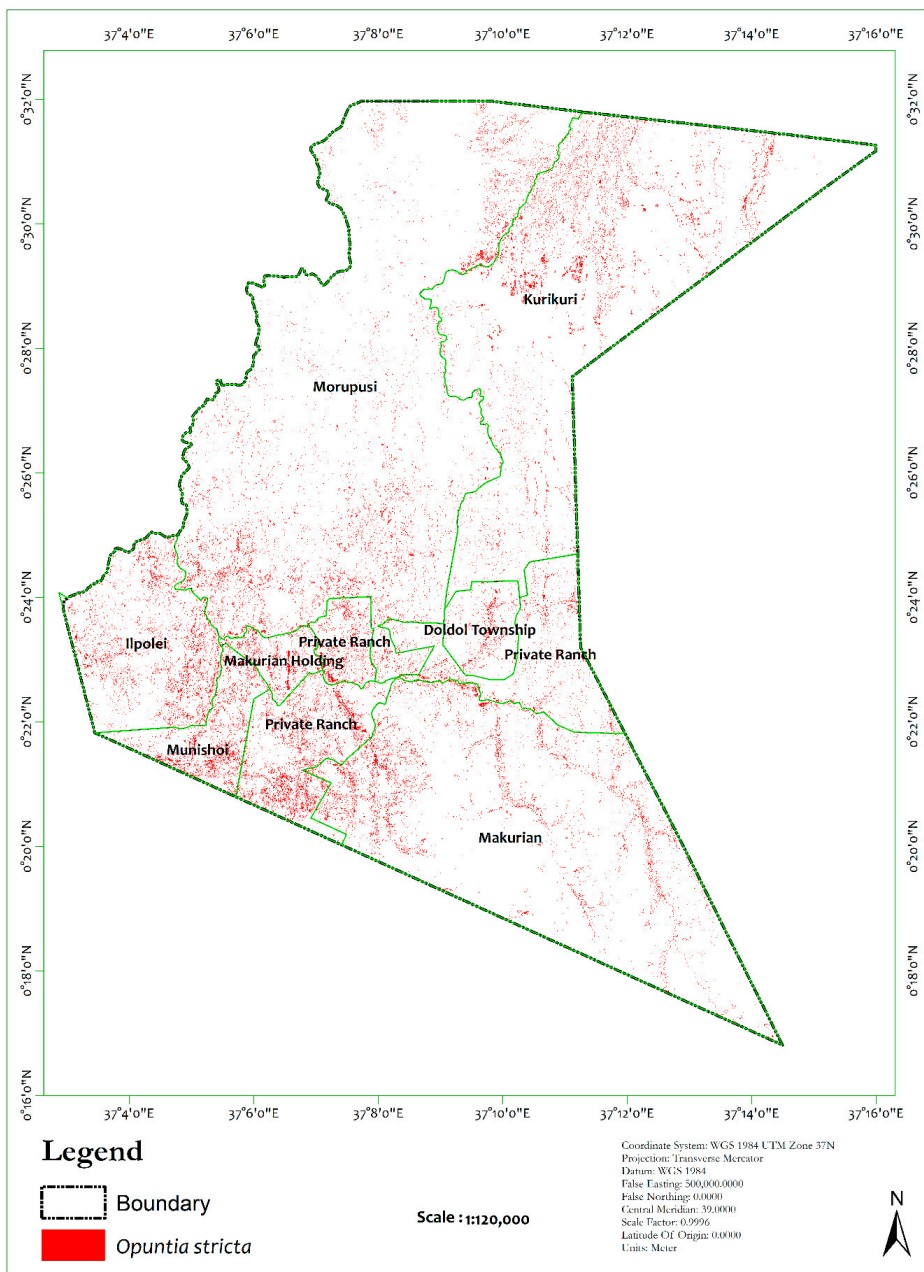

**Figure 7.** Map showing *Opuntia stricta* mask based on model 2.b.

## 4.2. Policy Implication, Limitation and Future Research

Our study can support policy decisions related to the control and management of *Opuntia stricta* in the ASALs. Our results have shown that riverine plains are heavily invaded thus restricting access to the 'last resort' forages and water, often relied upon by pastoralists during drought conditions. Additionally, the growth of *Opuntia stricta* in these areas results in more seed dispersal by water [23,33,62]. The continued land degradation [19,33] and the introduction of hard borders and fences [55,58] have led to competition for forage and water which can lead to conflicts [61]. This study informs pastoral management on (a) the restrictions of movement of their livestock in search for palatable pastures and (b) the possibility of encountering large abundances of *Opuntia stricta* in a specific region. Specifically, our study informs and contributes to policy [111] through the mapping and listing of threatened areas, research and development of invasive species as stipulated in the Kenya National Wildlife Strategy 2030 [112], addition of information to the already existing

rangeland monitoring web-based system to aid in the interpretation of the greenness metrics [113] and guides the application of control measures such as bio-control [114].

Our study identified limitations that are relevant for improved detection of *Opuntia stricta*. Firstly, our analysis was based on a relatively small sample that limits the detection to a localized small region. Changes in soil composition, management practices, and climate may alter the growth development of the IPS. Secondly, we found the target cover also grows under large canopies of riverine woody cover, hence these might not be captured by our method. Lastly, it is our understanding that a single pixel could represent a mixture of multiple end members hence we might not have captured all the variability that might exist in a single pixel. It is our considered view that future research addresses these identified limitations.

Future research could incorporate temporal aspects of plant phenology into the classification. This may be important as (a) flowering of plants may change spectral signatures seasonally, (b) the seasonal rain and dry periods will change the vegetation indices of grass and shrubby plants, and (c) livestock may graze or overgraze a location changing the vegetation mass and resulting spectral signatures. Furthermore, the application of an integrated (i.e., decision tree and mixed pixel decomposition) classification method using 3D terrain to improve the training data separability would be an interesting future contribution to the IPS detection. Finally, image processing techniques such as SAR polarimetry [115] enhance the mapping of *Opuntia stricta* especially in tropical regions that face cloud cover challenges limiting optical satellite use.

## 5. Conclusions

Our study sought to examine the ability of Sentinel-2 spectral and spatial properties to detect *Opuntia stricta* based on ensemble machine learning classifiers. This was the first time that *Opuntia stricta* mapping has been conducted based on satellite observations hence providing novel insights into its classification especially in heterogeneous ASALs. We also showed that spectral and topographic indices can meaningfully improve detection of *Opuntia stricta* and overall characterize complex heterogeneous ASALs with very good overall accuracies and Kappa, i.e., 89%, 92% and 0.87, 0.91 for XGBoost and RF classifiers, respectively. Comparison of accuracies of the ensemble machine learning classifiers signifies their ability to detect *Opuntia stricta* and other control classes. We demonstrated that incorporating spectra with topographical indices improved the overall accuracy. Finally, our work contributes to conservation and rangeland management policies that aim to map and list threatened areas, and conserve the biodiversity and productivity of rangeland ecosystem system.

**Supplementary Materials:** The following are available online at https://www.mdpi.com/article/10.3390/rs13081494/s1, Table S1: vegetation indices, Table S2: topographic indices, Table S3: results of the Shapiro-Wilk normality test, Figure S1: feature importance analysis.

**Author Contributions:** Conceptualization, J.M.M.; methodology, J.M.M.; software, J.M.M., E.E.S. and Z.-F.Y.; validation, J.M.M. and E.O.; formal analysis, J.M.M.; investigation, J.M.M.; writing—original draft preparation, J.M.M.; writing—review and editing, J.M.M., P.R. and A.S.A.; visualization, J.M.M.; supervision, P.R. and A.S.A. All authors have read and agreed to the published version of the manuscript.

**Funding:** This research was funded by the NERC Science for Humanitarian Emergencies and Resilience Studentship Cohort (SHEAR SSC) grant number: NE/R007799/1, and the SHEAR ForPAc project grant number: NE/P000673/1.

**Institutional Review Board Statement:** Not applicable.

**Informed Consent Statement:** Not applicable.

**Data Availability Statement:** Data to this article can be found online at https://doi.org/10.25377/sussex.14332718 (accessed on 10 March 2021).

**Acknowledgments:** Our appreciation to the Regional Centre for Mapping of Resources for Development for providing the validation data. We are also grateful to Margaret Wambua for guidance during the data collection. Furthermore, we thank the two anonymous reviewers for their constructive comments on this paper.

**Conflicts of Interest:** The authors declare no conflict of interest.

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
