# Peer review of "Mapping Opuntia stricta in the Arid and Semi-Arid Environment of Kenya Using Sentinel-2 Imagery and Ensemble Machine Learning Classifiers"

_remotesensing, doi:10.3390/rs13081494_

Round 1

Reviewer 1 Report

First version has been reviewed in September 2020. This new version presents many improvements and corrections and can be accepted in its present state.

One mistake should be corrected : Figure 6 appears twice. Opuntia mask should be labelled Figure 7 and mentioned in the text.

line 446 should be corrected accordingly:  Figures 5, 6 and 7 (instead of "Figure 5 and 8")

Author Response

We appreciate Reviewer 1 for taking the time to read through our manuscript and for taking note of the improvements made since the last submission. It is the invaluable feedback from reviewers, like yours, that have strengthened our paper and we would like to thank you for that. Below attached we have provided detailed responses to all the errors that you have identified.

Reviewer 2 Report

Mapping Opuntia stricta in the arid and semi-arid 3 environment of Kenya using Sentinel 2 imagery and 4 ensemble machine learning classifiers.

Remote Sensing- 1160058

This article uses the Sentinel-2 multispectral sensor to detect Opuntia stricta in a heterogeneous 21 ASAL in Laikipia County, using ensemble machine learning classifiers in order to identify the critical invaded plant species (IPS) in the ASAL landscape. Insights of the distribution of such in IPS is critical to formulate the appropriate management options in such critical landscape. The presented results are relevant for forest professionals and policy makers alike.

Title: Title is fine.

Abstract: Abstract seems a bit not crisp. Please make a clear especially after results.

Key words: Fine.

Introduction: The flow and intend of the first paragraph is not clear here. For example you mention “Ecologically, grasslands provide services for carbon sinks [1,2], biodiversity conservation [3], and forage biomass for herbivores [4,5]”. But there are other functions such as soil conservation. Therefore, be specific and accurate. You can refer to:

  1. Global trend of forest ecosystem services valuation–An analysis of publications
  2. Local users and other stakeholders' perceptions of the identification and prioritization of ecosystem services in fragile mountains: a case study of Chure Region of 
  3. An Ecosystem Services Valuation Research Framework for Policy Integration in Developing Countries: A Case Study from Nepal

For the background of the paper.

Methodology: Study area: Please write some background information on the ASAL.

Results: Good.

Discussion: Please make a sub-heading in the discussion so that it is easy to comprehend the discussion. Please refer and provide policy implication as of journal article 2.

Conclusion: Looks fine.

Author Response

We appreciate and thank Reviewer 2 for making time and thoroughly reading our manuscript as well as providing constructive feedback. Below attached, we have provided responses to the comments and we hope we have adequately responded to all your concerns.

Reviewer 3 Report

See attached file.

Author Response

We appreciate and thank Reviewer 3 for making time and thoroughly reading our manuscript as well as providing constructive feedback. Below attached, find our response to all the comments raised

This manuscript is a resubmission of an earlier submission. The following is a list of the peer review reports and author responses from that submission.

Round 1

Reviewer 1 Report

This research is about invasive species mapping based on Sentinel multispectral images. I believe this topic has significant research value because invasive species reduces biodiversity and threat native species in ecological concepts, and mapping for single species is always a challenge for remote sensing application. However, I have some comments and suggestions which may help improve the manuscript in current format.

For the introduction: literature review for remote sensing applications on species mapping is weak. The summary of remote sensing methods? Strength and weakness? What are the research gaps?

For Method part: The cartography of Figure 2 really needs to improve. The labels are all not clear; The legend show “training” and “validation”, but there is no such things in the map. No need to have both scale bar and Scale: 1:150,000 together; I suggest to move description context in the map into figure caption.

For the results part: For Figure 3, Grasses have similar spectral characteristics as Bare/Sand, which needs further explanation; Opuntia also has similar spectral signal with shrubs, which would cause lots of misclassification.

Reviewer 2 Report

see attached file

Reviewer 3 Report

The manuscript maps out the Invasive Plant Species (IPS) in a typical area in Kenya using a machine learning method. Grasslands provide a significant use and non-use values globally, and therefore the research has the potential to make a significant contribution to the current literature on grassland ecosystem functions. The manuscript is well organized with relatively few typographical and grammatical errors and a thorough description of the data. It also addresses a highly relevant issue in the grassland ecosystem function literature.

However, the manuscript needs some improvements in the following areas:

Title: Quite good and well

Key-words: Repeated with title such as Opuntia stricta, machine learning etc. 

Do not repeat these in key-words. 

Abstract: Well-written, however, there is some confusion. What is policy use of your is missing rather more advertising to describe the Sentinel-2 multispectral sensors? Make a balance and write few lines what is the policy implication of this study?

Introduction: Well-organised but there are many grammatical errors such in  L38-39 (Globally, grassland biomes form one of the largest covers of the terrestrial ecosystems and plays a critical roles ecologically, socially and economically.first line you use). 

In L38 delete the word good. 

In L45: what are other challenges, mention few of them.

In L48-L58 you briefed about the IPS in Africa and Kenya. Make a few lines about the research situation on IPS in your area and justify the research need here. 

In L75 you stated cultural measures? What does it mean here? 

Methodology: 

InL154-L156 you stated about the study area selection: Could you briefly outline why you choose this area for the study? 

Field data: L193: What is field campaigns? What is the sampling intensity and what is the basis of sampling and how you come up a total number of samples 628? 

Results:

3.1. you mentioned Similarly, we also found the weakest separation to be between Opuntia stricta and shrubs, 319 and bare grounds versus grasses, with values of JM equal to 1.27 and 1.49 respectively (Figure 3). Where is 1.27 and 1.49 in the figure?

3.2 Image classification. You mentioned many things here and get confused. You can separate them and make a clear. Explain only main things or your area of interest here and do not mix all in one.

Discussion: This section lacks the direct link to results section. Can you please add a discussion section corresponding to results, and discuss especially what is the primary trend/results, compare results with global or regional literature. Add discussion with why this type of trend exhibits in details.

Conclusion: What is the policy implications were missing in your conclusion section. Add a few potential policy benefits in conclusion section.